# Identification of Three *Epichloë* Endophytes from *Hordeum bogdanii* Wilensky in China

**DOI:** 10.3390/jof8090928

**Published:** 2022-08-31

**Authors:** Tian Wang, Taixiang Chen, James F. White, Chunjie Li

**Affiliations:** 1State Key Laboratory of Grassland Agro-Ecosystems, Key Laboratory of Grassland Livestock Industry Innovation, Ministry of Agriculture and Rural Affairs, Engineering Research Center of Grassland Industry (Ministry of Education), Gansu Tech Innovation Centre of Western China Grassland Industry, Center for Grassland Microbiome, College of Pastoral Agriculture Science and Technology, Lanzhou University, Lanzhou 730000, China; 2State Key Laboratory of Plateau Ecology and Agriculture, Qinghai University, Xining 810000, China; 3Department of Plant Biology, Rutgers University, New Brunswick, NJ 08901, USA

**Keywords:** *Epichloë*, *Epichloë* sp. HboTG-2, *Hordeum bogdanii* Wilensky, peramine, vertical transmission

## Abstract

Cool season grasses often form reciprocal symbiotic relationships with endophytic fungal species in genus *Epichloë*. In this study, we characterized three fungal endophytes isolated from the grass *Hordeum bogdanii* native to northwest China. Based on morphological characteristics and phylogenetic analyses of *tefA*, *tubB*, and *actG* sequences, we identified them as *Epichloë* sp. HboTG-2 (*H. bogdanii* Taxonomic Group 2: *E. bromicola* × *E. typhina*). Alkaloid synthesis related genes analysis showed that *Epichloë* sp. HboTG-2 may have the ability only to produce peramine which is toxic to insects but not to animals. In the process of this study, we did not observe sexual structures or epiphyllous growth on leaves of infected plants.

## 1. Introduction

Fungal endophytes of the genus *Epichloë* are organisms whose mycelium does not enter the host grass tissues and only grows in their intercellular spaces with a mostly unbranched morphology [1]. Many types of cool-season grasses can form a mutually beneficial symbiosis with *Epichloë* species [2,3]. Different from parasitism, mutualism reflects the fact that *Epichloë* species evolve synergistically with their hosts, do not cause overt symptoms in their hosts, and can provide the host with certain benefits, increasing the advantages of host grasses in ecological competition. For instance, *Epichloë* is known to supply host grasses with resistance not only to abiotic stresses such as drought, salinity, heavy metal [4,5], and low temperatures [6], but also to biotic stresses such as those from pathogens [7] and insect pests, whilst in return, the hosts offer a source of nutrients for the growth of *Epichloë* [8]. To untangle the mechanisms of *Epichloë* endophytes in order to help host grasses against various stressors is quite complex; at least, the view of co-evolution can help us understand why *Epichloë* species can establish particular mutualistic associations with many cool-season grasses: in their long evolutionary history, for each other’s survival and mutual interests, the *Epichloë* has made a gradual change from pathogen to mutualist [9]. Not all *Epichloë* species, however, can provide their host with benefits, and in some cases they do exactly the opposite, i.e., while these kinds of *Epichloë* species bring some benefits to the host, they also show negative impacts. For example, certain species can form a sexual structure (stroma) in their host that inhibits seed production, which is commonly known as “choke disease” [10]. These sexual species of *Epichloë* mainly transmit horizontally, but most of the *Epichloë* genus rarely or never form structures or ascospores related to horizontal transmission, instead spreading vertically through the seeds [11]. This mode is beneficial to the host, as it is based on the ability of *Epichloë* to colonize systemically within the plant intercellular tissues, infiltrating them with toxic compounds to herbivorous organisms, and to colonize seeds in order to allow the protection that *Epichloë* bestows upon it to be passed down steadily from generation to generation via vertical transmission [12].

In the process of establishing a stable symbiotic relationship between grasses and *Epichloë*, the ability of *Epichloë* to produce alkaloids to provide the host with direct defense against external damage, such as that from herbivores, is also extremely important [13]. A well-performing symbiotic association between *Epichloë* and grasses showed a complex cellular and molecular dialogue that often leads to the accumulation of bioprotective alkaloids results from improved resource allocation [14]. There are four main types of alkaloids produced by grass-*Epichloë* symbiota: pyrrolopyrazine, ergots, indole-diterpenes, and saturated pyrrolizidine, whose corresponding typical representatives are peramine, ergonorine, lolitrem B, and loline [15]. Peramine is the most common alkaloid in grass-fungal associations, and the simplest one of the four major alkaloids because it is not majorly toxic to livestock, but is very effective in fending off certain insects, especially *Listronotus bonariensis*, from feeding on hosts [16]. Loline is similar to peramine, being highly toxic to insects, but seems to have a certain negative impact on livestock at the same time. However, several recent studies have shown that the level of harm caused in this respect is nominal and can be ignored [17]. Ergots and indole-diterpenes are more toxic and can harm both incoming insects and herbivores: ergots are a group of nitrogenous compounds which come from L-tryptophan [18], the toxicity of ergots to livestock is achieved through vasomotor effects and influencing plasma prolactin concentrations [19]; and if livestock consumes grasses containing lolitrem B, the symptoms of toxicity are varied, ranging from light shivering of the neck to severe tetanic spasm and collapse, in terms of a response to these toxic effects, studies have shown that inhibition of the large-conductance calcium-activated potassium channels (BK channels) exploited by lolitrem B seems to be the only possible mechanism for detoxification [20]. Notably, there are no known endophytic fungal strains that can produce all four of these alkaloids at the same time, but one, two, or three of them can, which creates multiple combinations of alkaloid production between different endophytic fungal isolates in different grass-*Epichloë* symbiotic relationships. At present, the key cluster genes related to the synthesis of these four major groups of alkaloids above are known, meaning we can easily predict the alkaloid-producing types of *Epichloë* isolates via the polymerase chain reaction (PCR) method to screen out the isolates needed [21], in other words, the *Epichloë* isolates that do not contain toxic alkaloids in theory which may help in the establishment of sustainable agricultural practices by building up host grass fitness via production of defensive alkaloids [14].

The *Hordeum* genus of grasses is composed of species that are important as food crops and for animal husbandry, and they serve as host grasses for *Epichloë* endophytes. The presence of *Epichloë* species in *Hordeum* was first reported in the early 1990s [22], but much more recently, novel barley germplasm has been created using *Epichlo*ë endophytes of *E. bromicola* from *Hordeum brevisubulatum* [23], highlighting the research progress made in *Epichloë* endophytes from *Hordeum* plants. *Hordeum bogdanii* is a kind of salt-tolerant plant resource with strong ecological adaptability and competitive advantages, widely distributed in northwest China. In the aforementioned study by Wilson et al. [22], *H. bogdanii* infected with *Epichloë* was first discovered in the U.S.A few years later, it was discovered in the Altay region of Xinjiang, China, and the infection rate of seeds was reported to be 18% [24]. Besides, compared with *H. bogdanii* without *Epichloë*, the presence of *Epichloë* was found to promote growth measures of *H. bogdanii* such as total biomass, dry weight, and the number of tillers [25]. Up to now, four different *Epichloë* species (including three unnamed taxonomic groups) have been identified from *H. bogdanii* worldwide. The northwest region of China is the main area where *H. bogdanii* is distributed, but there is a lack of systematic research on the classification status and characteristics of alkaloid production of *Epichloë* within *H. bogdanii* in China.

In this study, we re-collected samples of *H. bogdanii* from parts of north China (mainly in Xinjiang) and isolated the endophytic fungi strains of *Epichloë*. Then, in addition to describing its morphological characteristics, we performed a phylogenetic analysis of the isolated strains and PCR analysis of alkaloid synthesis genes to assess their potential for alkaloid production. The aim was to screen animal-safe grass-endophyte isolates that are toxic only to insects and nontoxic to livestock, thereby providing a theoretical basis and fundamental materials for subsequent research and applications of *Epichloë* in breeding work related to forage or cereal crops.

## 2. Materials and Methods

### 2.1. Biological Materials and Isolation of Endophytes

Plant materials with mature seeds were collected at an altitude of 1077 m from Yumin county (82°53′21″ N, 46°3′37″ E; Xinjiang, China). After the materials were brought back to the laboratory, the endophyte infection rate was immediately determined microscopically in stalk tissues stained for fungal hyphae with aniline blue [26]. The seeds of plants were stored at 4 °C to maintain the viability of endophytic fungi, and some seeds were also sown in the greenhouse (18–24 °C), and watered as required for seed multiplication, in order to facilitate subsequent trials and an adequate supply of trial material.

The seeds that looked healthy were used to isolate the fungal endophyte of *Epichloë*. First, the surfaces of the seeds were disinfected (70% absolute ethanol for 3 min, 5% sodium hypochlorite solution for 3 min), and then washed three times with sterile distilled water. Next, the seeds were completely dried with sterilized filter paper, placed on PDA (potato dextrose agar) with 100 μg/mL of ampicillin and 50 μg/mL of streptomycin sulfate, sealed with parafilm, cultured in the dark at 22 °C, and examined regularly for endophyte growth for up to 2 weeks. Endophyte colonies growing out from tissues were transferred to fresh media, and all isolates were deposited at the Mycological Herbarium of Lanzhou University, Lanzhou, China.

### 2.2. Morphological Examination

The colony morphology of three isolates was examined from cultures grown on PDA. Mycelial discs, 0.4 cm in diameter, were inoculated on PDA at 22 °C in the dark, and for each isolate, we set up six repeats. From cultures grown for 32 days, the colony diameter rate was measured, the radial growth rate recorded and averaged, the cultures photographed, and the colony morphology of the three isolates described. The conidia and conidiogenous cells were photographed and measured from mycelium grown on 1.5% WA (water agar) plates. The 0.4-cm mycelial discs were inoculated on the WA plates at 22 °C in the dark for 4 weeks, and the size of the conidia was measured (width and length, *n* = 50 respectively) along with the conidiogenous cells (length, *n* = 30), with an automated upright fluorescence microscope (Olympus Corporation, Tokyo, Japan, Olympus, BX63). The data are shown as mean ± standard error.

### 2.3. DNA Extraction

After growth for 2 weeks on PDA at 22 °C, several pieces of fresh mycelial discs were inoculated in PDB (potato dextrose broth) and shaken at 22 °C at 120 rpm for three weeks, and then the mycelia were collected into Eppendorf tubes. Total genomic DNA was extracted using an HP fungal DNA kit (OMEGA, Beijing, China) according to the manufacturer’s instructions.

### 2.4. PCR Amplification of the Housekeeping Genes tubB, tefA, and actG of Three Isolates and Purification of PCR Products

Three housekeeping genes were selected for PCR amplification of DNA from three isolates: the portions of the β-tubulin (*tubB*) gene, translation elongation factor 1-alpha (*tefA*) gene, and actin (*actG*) gene. The primers used were tub2-exon 4u-2 (GTTTCGTCCGAGTTCTCGAC), tub2-exon 1d-1 (GAGAAAATGCGTGAGATTGT), tef1-exon 5u-1 (CGGCAGCGATAATCAGGATAG), tef1-exon 1d-1 (GGGTAAGGACGAAAAGACTCA), act1-exon 6u-1 (AACCACCGATCCAGACAGAGT), and act1-exon 5u-1 (TAATCAGTCACATGGAGGGT) [27,28]. PCR was carried out in a total volume of 25 μL. The PCR cycling conditions were an initial denaturation step: of 5 min at 94 °C, then 30 cycles of denaturation at 94 °C for 30 s, annealing for 45 s at 45 °C (*tubB*), 55 °C (*tefA*), or 50 °C (*actG*), a 72 °C extension for 1 min, and a final step at 72 °C for 10 min, before being stored at 4 °C. Amplified products were analyzed by agarose gel electrophoresis with 1.5% agarose gel in 1 × TAE buffer. Sanger sequencing was commissioned at Bioengineering (Shanghai, China) Co. The sequencing results of the PCR products from the three isolates were found to be bimodal, so they were further purified. The PCR products were detected by 1.5% agarose gel electrophoresis and the target gene fragments were cut by blade under UV irradiation and purified by a DNA purification kit. The purified products were cloned into the pGEM-T Easy vector (Promega) following the manufacturer’s instructions. For each of the three isolates, 10 positive clones were randomly selected for sequencing by Bioengineering (Shanghai) Co. Unique sequences were submitted to GenBank. [Accession numbers were: *actG* (MW923963, MW923964, MW923965, MW923969, MW923970, and MW923976); *tefA* (MW923987, MW923988, MW923989, MW923990, and MW923996); *tubB* (MW924016, MW924007, MW924008, and MW924009)].

### 2.5. Phylogenetic Analysis

The sequencing results were entered into GenBank to detect whether they belonged to the genus *Epichloë*. If they did, they were compared with other known *Epichloë* species in MAFFT 7.037 [29], a selection of conserved sequences, using the online software Phylogeny.fr: Gblocks 0.91b [30]. Sequence saturation detection was then carried out using the software DAMBE. Selection of optimum model and construction of phylogenetic trees (ML) using the IQtree 2.2.0 with setting the bootstrap value to 1000 by using 1000 random repetitions of sampling. The maximum parsimony tree was constructed by PAUP 4.0 software.

### 2.6. Alkaloid Gene Detection

To test which alkaloid synthesis and mating-type genes were contained in these three isolates, PCR was applied to the total genomic DNA of the three isolates using the target-specific primers previously described by Charlton et al. and Berry et al. [31,32]. The genes tested included 7 fragments of the *perA* gene in peramine biosynthesis, 14 key *EAS* cluster genes of ergot alkaloids, 11 *LOL*, and 11 *LTM*/*IDT* cluster genes in the biosynthesis of loline and indole-diterpenes alkaloids. The PCR cycling parameters were 94 °C for 1 min, followed by 30 cycles of 94 °C for 15 s, 56 °C for 30 s, 72 °C for 1 min, and then 10 min at 72 °C, before a final annealing temperature of 4 °C. Products were analyzed in 1 × TAE buffer by gel electrophoresis with agarose gels of 1.5%. We selected *E. inebrians* E818 and *Epichloë* sp. FS001 as positive controls for genes at the *EAS* locus and lolitrem B [33,34], respectively. As positive controls for *perA* genes and genes at the *IDT* locus, we used *E. festucae* var. *lolii* AR1 [35,36].

## 3. Results

### 3.1. Endophyte Characteristics

A total of three isolates (D2-1-H, D2-2-B, D2-1-B) were isolated from surface-sterilized seeds of *H. bogdanii* grasses, and they all conformed to the typical characteristics of *Epichloë* species. In culture on PDA, the mycelium of D2-1-H was sparse, white to earthy yellow, lightly felted, and surrounded by a dark growth circle. The back of the colony was dark brown at the center, and the color gradually became lighter towards the edge of the colony. The outermost circle had a white aperture and was slow growing (Figure 1A); the average growth was 0.62 mm per day in 32 days (Table 1). In colonies of D2-2-B, the aerial mycelium was dense, raised, and formed pure white cotton-like colonies surrounded by a light brown to dark brown growth circle. The mycelium above the growth circle was sparse (Figure 1G,H). The aerial mycelium of D2-1-B (Figure 1M) was dense and formed a cotton-like colony. The central mycelium was tight and flat. The surrounding growth circle was sparser than the center, and the color was lighter than the other two isolates but the mycelium was denser. The color of the back of the colony was light brown, and the growth rate of D2-1-B was almost the same as that of D2-2-B (Table 1). Conidiogenous cells of the three *Epichloë* isolates showed the same appearance, being phialidic, sparse, arising solitarily and perpendicularly from aerial hyphae, hyaline, and 6.27–29.12 µm long (Figure 1C–F,I–K,N–P). The conidia of the three strains showed the same characteristics too, being reniform or asymmetric, hyaline, and 3.48–7.07 × 1.19–3.39 μm (Figure 1C–F,I–K,N–P). Compared with previous reports, the growth rates of the three isolates in this study were significantly slower or faster than some *Epichloë* strains (Table 1), such as *E. elymi* (*Elymus* sp.), *E. glyceriae* (*Glyceria striata*), *E. aotearoae* (*Echinopogon ovatus*), and *E. uncinate* (*Lolium pretense*) [27,37,38]; however, the growth rates of some *Epichloë* species, such as *E. typhina* subsp. *clarkia* (*Holcus lanatus*), was similar to those of the isolates in this study, but their conidia and conidiogenous cells were significantly different [39]. For brevity, we do not list in detail the morphological differences between the three isolates in this study and all other known *Epichloë* species (more than 40 taxa). Rather, we illustrate some key examples by comparing their morpho-physiological differences, which we preliminary to distinguish between the three isolates obtained in this study from other known *Epichloë* species. For example, compared with reported *Epichloë* isolates from *Hordeum* plants (Table 1), the growth rates of all three isolates in our study were lower than that of *E. bromicola* (*H. europaeus*) but faster than that of *E. tembladerae* (0.5 mm/day, *H. comosum*). The growth rate of *E. bromicola* (1.21 mm/day, *H. bogdanii*) was similar to that of D2-2-B, but the *E. bromicola*, length of its conidiogenous cells, and width of its conidia were longer than those of D2-2-B. However, in culture, the morphological indexes were easily affected by culture conditions, and the taxonomic status of the three strains could not be accurately identified from morphological observation.

### 3.2. Phylogenetic Analyses

Two different gene homeologs of *H. bogdanii* endophytes (isolates D2-1-H, D2-2-B, and D2-1-B) were identified from the PCR amplification of *tefA*, *tubB*, and *actG*, indicating that all three isolates were hybrids with two alleles. Regardless of their morphological differences, the three isolates had the same taxonomic position in the maximum likelihood phylogenetic tree (Figure 2, Figure 3 and Figure 4) and maximum parsimony (MP) tree (Figure 5, Figure 6 and Figure 7), including two alleles, and these two alleles were in different subclades of the *Epichloë* tree. In the phylogenetic tree of *tefA*, *tubB*, and *actG*, allele 1 of three isolates grouped with *E. bromicola*, allele 2 grouped with *E. typhina*.

From the above results, allele 1 and allele 2 were grouped with known *Epichloë* species—*E. bromicola* and *E. typhina*, respectively. This constitutes the molecular evidence that the three isolates in this study belong to *Epichloë* sp. HboTG-2.

### 3.3. Mating Types and Alkaloid Gene Profiling

The total DNA of all three *H. bogdanii* isolates was tested by PCR for their mating type and the presence of the key genes for alkaloid production. The results indicated that the three isolates (D2-1-H, D2-2-B, and D2-1-B) were consistent in mating type and alkaloid production profiles. They all contained both markers, *mtAC* and *mtBA*, which are for mating type *MTA* and *MTB* (Table 2), revealing them to be hybrids (also confirmed by phylogenetic analysis). The three isolates all had seven markers of the *perA* gene required for peramine synthesis, but lacked the *perA*-ΔR* allele (Table 2), a fragment detrimental to the synthesis of peramine, indicating that they could produce peramine. Of the 14 key genes tested from the *EAS* locus, three isolates contained fragments of *easF*, *easE*, *easD*, *easG*, *lpsA* and *lpsB*, but lacked the remaining eight genes, including *dmaW*—the first key gene in the *EAS* pathway; and because of the absence of this gene, no ergot-like alkaloids of any kind, including chanoclavine I (CC), were produced (Table 2). Of the 11 genes tested from the *LOL* locus, only *lolC* existed (Table 2). This means that not only are the three strains unable to produce loline, but also other products such as 1-acetamido-pyrrolizidine (AcAP), *N*-acetylnorloline (NANL), *N*-methylloline (NML), and *N*-formylloline (NFL). None of the three isolates contained any of the 11 key genes required for the synthesis of indole-diterpenes (Table 2), indicating that they cannot synthesize any of the products in the lolitrem B pathway.

**Table 1 jof-08-00928-t001:** Morpho-physiological comparison of *Epichloë* isolates in this study with selected *Epichloë* species.

Endophyte	Host	Growth on PDA (mm/day)	Length of Conidiogenous Cell (μm)	Conidia Size (μm)		Reference
				Length	Width	
D2-2-B	*Hordeum bogdanii*	1.20 ± 0.01a (22 °C)	12.81 ± 0.66 b	5.74 ± 0.97 a	2.48 ± 0.52 a	This study
D2-1-B	*Hordeum bogdanii*	1.13 ± 0.02 a (22 °C)	13.03 ± 0.56 b	5.54 ± 0.97 a	2.28 ± 0.53 b	This study
D2-1-H	*Hordeum bogdanii*	0.61 ± 0.04 b (22°C)	18.80 ± 1.02 a	4.96 ± 0.83 b	2.36 ± 0.43 ab	This study
*Epichloë* sp. HboTG-1	*Hordeum bogdanii*	nt	nt	nt	nt	[40]
*Epichloë* sp. HboTG-2	*Hordeum bogdanii*	0.83 ± 0.06	17.4 ± 6.2	5.9 ± 1	3.1 ± 0.5	[41]
*Epichloë* sp. HboTG-3	*Hordeum bogdanii*	0.99 ± 0.09	21.8 ± 5.7	7.4 ± 1	4.47 ± 0.4	[41]
*Epichloë* sp. HbrTG-1	*Hordeum brevisubulatum*	nt	nt	nt	nt	[40]
*Epichloë* sp. HbrTG-2	*Hordeum brevisubulatum*	nt	nt	nt	nt	[40]
*Epichloë tembladerae*	*Hordeum comosum*	0.5 (24 °C)	19.8 ± 0.73	8.7 ± 0.17	3.2 ± 0.09	[42]
	*Hordeum comosum*	0.94 (24 °C)	22.3 ± 1.9	8.5 ± 0.18	3.4 ± 0.15	[42]
*Epichloë bromicola*	*Hordelymus europaeus*	1.43–1.67 (24 °C)	20.2 ± 4.7	4.2 ± 0.4	2.1 ± 0.2	[43]
	*Hordeum brevisubulatum*	0.77 ± 0.02 (22 °C)	19.50 ± 1.06	5.17 ± 0.06	2.87 ± 0.17	[44]
	*Hordeum bogdanii*	1.21 ± 0.1 (23 °C)	14.0 ± 3.5	4.6 ± 0.4	2.7 ± 0.3	[41]
	*Hordeum bogdanii*	0.99 ± 0.1 (23 °C)	19.5 ± 5.7	5.0 ± 0.5	2.7 ± 0.3	[41]
*Epichloë elymi*	*Elymus* sp.	1.95–2.86 (24 °C)	17 ± 3	4.0 ± 0.4	2.2 ± 0.2	[37]
*Epichloë chisosa*	*Stipa eminens*	0.26 (20 °C)	10–30	5–9	2.5–4	[45]
*Epichloë typhina*	*Dactylis glomerata*	1.86–3 (25 °C)	13–33	4.1 ± 0.5	2.2 ± 0.5	[46]
*Epichloë typhina* subsp. *clarkii*	*Holcus lanatus*	1.22 (23~24 °C)	33.8 ± 6	4.4 ± 0.4	1.9 ± 0.1	[39]
*Epichloë aotearoae*	*Echinopogon ovatus*	<0.31 (22 °C)	nt	nt	nt	[27]
*Epichloë glyceriae*	*Glyceria striata*	2.48–3 (24 °C)	31 ± 5	3.8–6.2	2.2–2.8	[37]
*Epichloë brachyelytri*	*Brachyelytrum erectum*	0.76–2.43 (24 °C)	16 ± 3	4.1 ± 0.6	2.8 ± 0.3	[37]
*Epichloë baconii*	*Agrostis capillaris*	0.75 (23~24 °C)	24.8 ± 3.7	4.4 ± 0.6	1.75 ± 0.25	[39]
*Epichloë uncinata*	*Lolium pretense*	<0.43 (25 °C)	9–18	5–13	1–2	[38]
*Epichloë coenophialum*	*Festuca arundinacea*	0.29 (20 °C)	12–34	7–11	2–3	[47]
*Epichloë festucae*	*Festuca rnbra*	1–2.67 (24 °C)	12–25	4.7 ± 0.6	2.2 ± 0.3	[48]
*Epichloë novae-zelandiae*	*Poa matthewsii*	0.14–0.17 (24 °C)	14–40	3.3–5.7	1.8–3.1	[49]
*Epichloë scottii*	*Melica uniflora*	1.24–1.29 (20 °C)	14.1 ± 2.8	4.5 ± 0.4	3.0 ± 0.25	[50]

Different lower-case letters indicate significant differences among D2-2-B, D2-1-B, and D2-1-H (Duncan’s test, *p* < 0.05); nt: not tested

**Table 2 jof-08-00928-t002:** Alkaloid gene profiling to determine alkaloid chemotype classes.

Gene	Present or Absent in Endophyte Genome			Gene	Present or Absent in Endophyte Genome		
	D2-1-H	D2-2-B	D2-1-B		D2-1-H	D2-2-B	D2-1-B
**Mating-Type Genes**				**Loline (*LOL*) Genes**			
*mtAC*	+	+	+	*lolC*	+	+	+
*mtBA*	+	+	+	*lolF*	-	-	-
**Segments of *perA* Gene**				*lolD*	-	-	-
*PerA*-A1	+	+	+	*lolT*	-	-	-
*PerA*-T1	+	+	+	*lolA*	-	-	-
*PerA*-C	+	+	+	*lolU*	-	-	-
*PerA*-A2	+	+	+	*lolO*	-	-	-
*PerA*-M	+	+	+	*lolE*	-	-	-
*PerA*-T2	+	+	+	*lolN*	-	-	-
*PerA*-R*	+	+	+	*lolM*	-	-	-
*PerA-*ΔR*	-	-	-	*lolP*	-	-	-
**Ergot Alkaloid (*EAS*) Genes**				**Indole-Diterpene (*IDT*/*LTM*) Genes**			
*dmaW*	-	-	-	*idtG*	-	-	-
*easF*	+	+	+	*idtB*	-	-	-
*easE*	+	+	+	*idtM*	-	-	-
*easC*	-	-	-	*idtC*	-	-	-
*easD*	+	+	+	*idtS*	-	-	-
*easA*	-	-	-	*idtP*	-	-	-
*easG*	+	+	+	*idtO*	-	-	-
*cloA*	-	-	-	*idtF*	-	-	-
*lpsA*	+	+	+	*idtK*	-	-	-
*lpsB*	+	+	+	*idtE*	-	-	-
*easH*	*-*	*-*	*-*	*idtJ*	*-*	*-*	*-*
*lpsC*	-	-	-				
*easO*	-	-	-				

“+”: gene present in endophyte genome, “-”: gene absent from endophyte genome.

## 4. Discussion

The grasses of the genus *Hordeum*, belong to the family Poaceae, which is one of the most important cereal and forage crop families in China. Indeed, it has been planted for these purposes for thousands of years. In recent years, *Epichloë* isolates have been identified from several *Hordeum* plants. The earliest records are from 1991 when three *Hordeum* species, *H, bogdanii*, *H. brevisubulatum*, and *H. comosum*, were found to be infected by *Epichloë* species [22]; however, this was a brief report with information only about the percentage level of seed infection by the endophytic fungi and no further work on isolation and identification. According to the latest research, the animal-safe endophyte of *E. bromicola* strains isolated from *H. brevisubulatum* has been applied to the study of artificial inoculation for crop improvement [23]. Following further research on the *Epichloë* genus, at least nine different taxa (including unnamed taxonomic groups) have to date been identified from different *Hordeum* species. Among them, the *Epichloë* strains isolated from *H. bogdanii* include *E. bromicola*, *Epichloë* sp. HboTG-1 (*H. bogdanii* Taxonomic Group 1: *E. elymi* × *E. amarillans*), *Epichloë* sp. HboTG-2 (*H. bogdanii* Taxonomic Group 2: *E. bromicola* × *E. typhina*), and *Epichloë* sp. HboTG-3 (*H. bogdanii* Taxonomic Group 3: *E. bromicola* × *E. baconii*) [40,41]. In the present work, we isolated three fungal endophytes from *H. bogdanii* in Xinjiang, China, based on *tefA*, *tubB*, *actG* gene sequences, allele 1 of three isolates grouped with *E. bromicola*, allele 2 grouped with *E. typhina*, this same as *Epichloë* sp. HboTG-2, and, combined with the results of previous studies, we identified three isolates in our study as *Epichloë* sp. HboTG-2.

The genus *Epichloë* can protect its hosts to deter herbivores and insects by producing four major classes of alkaloids [51]. These alkaloids maintain polymorphisms in many *Epichloë* species, and sexual *Epichloë* species are more likely to mutate during transmission, meaning interspecific hybridization could pyramid the alkaloid biosynthesis genes. However, there are no known isolates that can produce all four types of alkaloids at the same time. To prevent endophytic fungi from consuming too many nutrients from the host, the host needs to prevent the production of some unnecessary alkaloids; that is, under the condition of these alkaloid gene clusters being regulated, the most advantageous interaction between the endophyte and host can be established [15]. For the survival of the host grasses, the existence of any type of alkaloid can provide them with a certain shelter effect; however, when screening for animal-safe endophytes, we target *Epichloë* isolates that do not produce alkaloids in the form of indole-diterpenes or ergot alkaloids. The non-toxic strains can be used in germplasm innovation for forage or crops. With improvement in our understanding of the gene clusters in the synthesis of these alkaloids, increasing emphasis has been placed on detecting, via PCR, which alkaloid biosynthesis genes are present within endophyte isolates, to evaluate the capability of *Epichloë* species to produce alkaloids [15,52]. According to the reported *Epichloë* species of *Hordeum*, they have the potential to produce the alkaloids peramine, ergovaline, terpendole E, terpendolend C, and *N*-formylloline (NFL) [41]. Among the reported *Epichloë* species from *H. bogdanii*, their potential for alkaloid production has been elucidated, except for *Epichloë* sp. HboTG-1. The *E. bromicola* from *H. bogdanii* has two different genotypes, which differ mainly in their ability to produce peramine, *Epichloë* sp. HboTG-3 has the potential to produce peramine, *N*-formylloline, and ergovaline [41]. In our study, three isolates belong to *Epichloë* sp. HboTG-2, can only produce peramine, because they lack eight *EAS* genes (*dmaW*, *easC*, *easA*, *cloA*, *easH*, *lpsC*, *easO*, and *easP*) among which *dmaW* is the first key gene in the *EAS* pathway. In addition, it lacks all of the *IDT* genes, and contain only the *lolC* gene of the *LOL* gene cluster, meaning they only produce peramine, which is toxic to insects but not to livestock, thus in theory making it an animal-safe endophyte. This result is in agreement with Yi et al. [41].

Would genetic variations (such as alkaloid synthesis genes) result from morphological variation in *Epichloë* species? It was found that three isolates of *Epichloë* sp. HboTG-2 in our study has some differences in morpho-physiological properties, in terms of growth rate, length of conidiogenous cells, and size of conidia, isolate D2-1-H differs significantly from isolates D2-1-B and D2-2-B. Compared with Yi’s work [41], the isolate NFe12 of *Epichloë* sp. HboTG-2 from *H. bogdanii*, also differs from three isolates obtained in this study in morpho-physiological properties. Additionally, we compared genes associated with alkaloid synthesis in the four isolates described above (D2-1-H, D2-1-B, D2-2-B, NFe12); they shared the same genes related to alkaloid synthesis, and their morpho-physiological variability did not result in genetic variation [41]. All four isolates mentioned above originated from Xinjiang, China, but host grass samples of isolate NFe12 were collected in 1983, while those of the three isolates in our study were collected in 2014. Additionally, additional isolates (NFe24, NFe16, and NFe39) of *Epichloë* sp. HboTG-2 identified from Kazakhstan (a country) maintain identical alkaloid-producing genes with the four isolates mentioned above [41]. As a result, despite morpho-physiological differences, *H. bogdanii* endophytes of *Epichloë* sp. HboTG-2 above from different years and countries retain the consistency of alkaloid biosynthesis genes. Is there a similar phenomenon in other *Epichloë* isolates from other *Hordeum* plants? Three isolates (WBE1, WBE3, and WBE4) of *Epichloë bromicola*. from *Hordeum brevisubulatum* [44], as compared to WBE1 and WBE4, WBE3 grew significantly faster at 22 °C and 25 °C, respectively. The taxonomic status and genes related to alkaloid synthesis are the same between them, and some morpho-physiological between *Epichloë* isolates in *Hordeum* may not cause genetic variations. In contrast, compared with other *Epichloë* isolates from other grasses, the situation is not always so. According to Charlton et al. [31], they obtained *Epichloë* strains from *Elymus canadensis* that had three morphotypes, all of which were identified as *Epichloë canadensis*. However, genetic variation was detected among the morphotypes for the *EAS* genes, there was also variation in alkaloids in different isolates of *E. elymi* [31]. Since the above studies did not identify sexual structures (stroma) or epiphyllous growth on leaves of infected plants, vertical transmission is likely to be the mode of transmission. Interestingly, the vertical transmission may retain relatively high genetic stability in *Epichloë* endophytes from *Hordeum* plants. It is well known that the defensive alkaloids that *Epichloë* provides to the host grasses can be passed down from generation to generation, largely due to the vertical transmission of *Epichloë*; however, from the above comparison, the genetic stability caused by vertical transmission seems to be different among different grasses or *Epichloë* species. In some *Epichloë* species, morphological variations lead to genetic variations mainly in the synthesis of alkaloids. Why and how this genetic variation caused by morphological variations occur among the same *Epichloë* species may be an interesting and meaningful point to be studied.

Hybrid *Epichloë* species with more copies often have multiple *EAS*, *IDT*, or *LOL* clusters which could lead to increased benefits to the host grasses, especially in their ability to produce alkaloids; they carry more genome genes, and may produce more diversified alkaloids than a haploid genome [53,54], for which there is recent empirical evidence [55]. According to our results, three isolates of *Epichloë* sp. HboTG-2 with two copies in our study can only produce peramine; it cannot produce ergot, or indole diterpenoid alkaloids. Another two-copy hybrid, *Epichloë* sp. HboTG-3, has the potential to produce peramine, *N*-formylloline, and ergovaline, because it is possession of more alkaloid synthesis genes [41]. This is somewhat similar to *E. novae-zelandiae*, a three-copy hybrid, in addition to producing peramine, can only produce some early products of ergot alkaloids, indole diterpenoids, or the loline alkaloid synthesis pathway, which are less toxic to herbivores. Based on the examples above. we would like to point out that *Epichloë* hybrids with more copies actually have the potential to synthesize more kinds of alkaloids, but the actual quantity of alkaloid production is also directly related to their parents’ genes [56]. If their parents do not possess certain genes, their hybrids are also unlikely to have those genes. Of course, the possibility of somatic mutations cannot be ruled out; and we believe it is this diverse hybridization and variety of changes that, whilst challenging, is appealing to study. In summary, it thus appears that hybrids with more gene copies have a greater potential to produce more diverse types of alkaloids; however, in practice, this potential is not always realized to its fullest, because some isolates lack some key genes required for certain pathway steps [53], as seen in this study. However, this is not to deny the potential for alkaloid synthesis in non-hybrid species; some single-copy *Epichloë* species also have multiple alkaloid synthesis genes, such as the endophytic fungal species of *E. bromicola* isolated from wild barley (*H. brevisubulatum*) discovered by Chen et al. [44].

In recent years, with the increasing limitations placed on the use of synthetic chemicals and the promotion of environmentally friendly agriculture, the application of animal-safe endophytes of *Epichloë* for breeding new grass varieties has become an international trend in the grass industry, and some new grass species have been successfully commercialized [57]. In theory, the three isolates obtained in this study, which can only produce peramine which is toxic to insects but not to animals, can be used in generating “endophyte engineered crops”. Some progress has been made in the study of germplasm innovation using *Epichloë* isolates from *Hordeum* plants. Li et al. [23] took an animal-safe strain WBE1 of *E. bromicola* isolated from *H. brevisubulatum*, and inoculated it into *H. vulgare* artificially, and successfully created a novel barley germplasm. In addition, TePaske et al. [58] found no livestock poisoning in a natural grazing experiment of several species of *Hordeum* plants infected by *Epichloë* in the early stage, thus revealing the considerable potential of *Epichloë* in *Hordeum* plants. However, we know that the use of *Epichloë* isolates for germplasm innovation is a complex task, because the transfer of endophytic fungi between ecotypes or species established artificially may lead to significant changes in their symbiotic relationships [59]. However, it is precisely because of this uncertainty that we need to conduct a large number of studies to know the signaling and molecular cascades between *Epichloë* endophytes and host grasses [14]. Such efforts will help us to improve our understanding of the symbiotic relationships between endophytic fungi and hosts.

## Figures and Tables

**Figure 1 jof-08-00928-f001:**
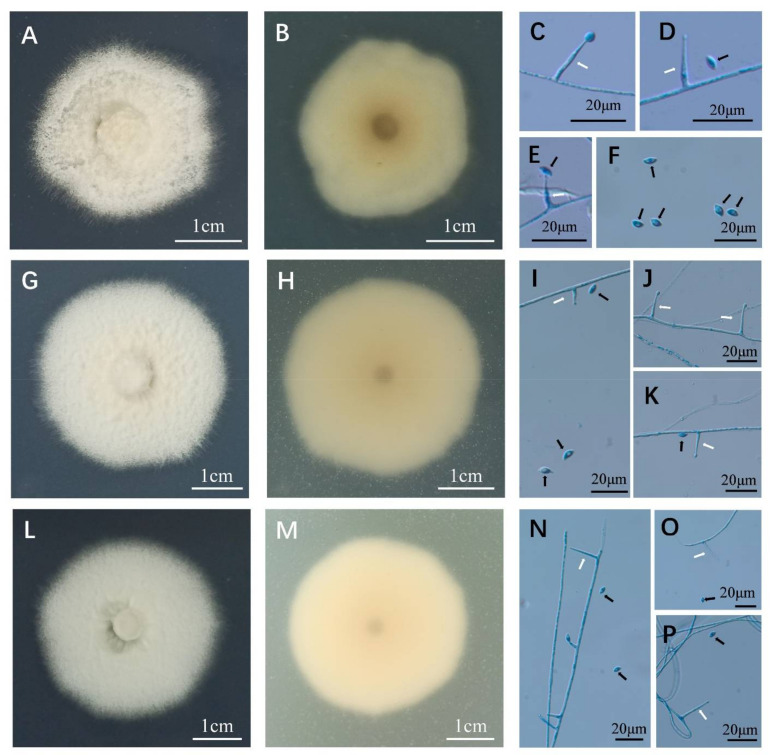
Colony morphology, and conidia and conidiogenous cells of *Epichloë* sp. HboTG-2 isolates from *H. bogdanii*. The colony is from cultures grown on PDA at 22 °C for 32 days. (**A**), the surface of D2-1-H; (**B**), the reverse of D2-1-H; (**C**–**F**), conidia (black arrow) and conidiogenous cells (white arrow) of D2-1-H; (**G**), the surface of D2-2-B; (**H**), the reverse of D2-2-B; (**I**–**K**), conidia (black arrow) and conidiogenous cells (white arrow) of D2-2-B; (**L**), the surface of D2-1-B; (**M**), the reverse of D2-1-B; (**N**–**P**), conidia (black arrow) and conidiogenous cells (white arrow) of D2-1-B. The conidia and conidiogenous cells are blue because they were stained with aniline blue solution.

**Figure 2 jof-08-00928-f002:**
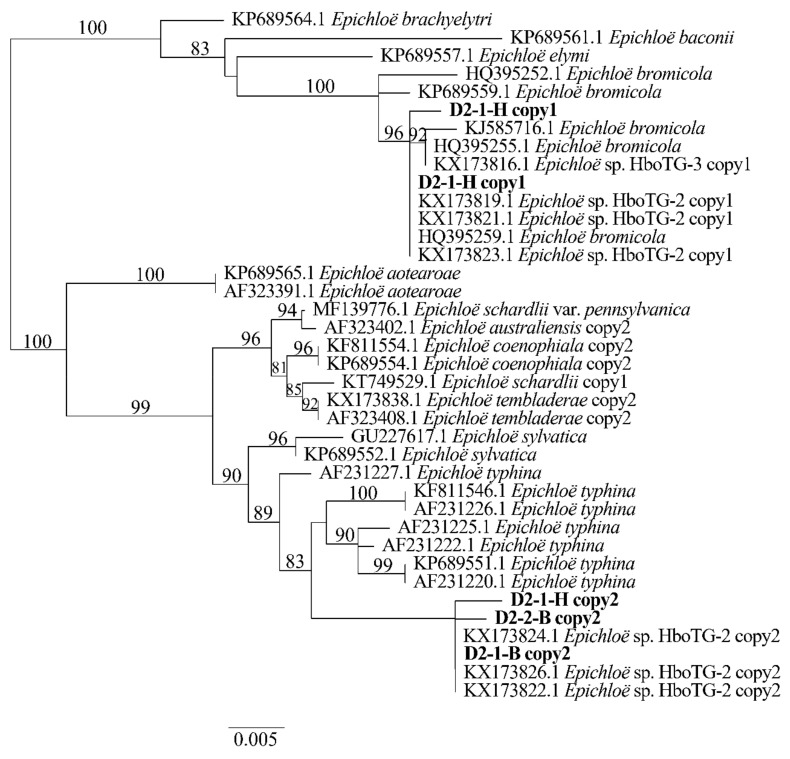
Molecular phylogeny derived from maximum likelihood (substitution model TIM3e + G4) analysis of the *tefA* gene from representative *Epichloë* species and copies of the hybridized endophytes isolated from *H. bogdanii*.

**Figure 3 jof-08-00928-f003:**
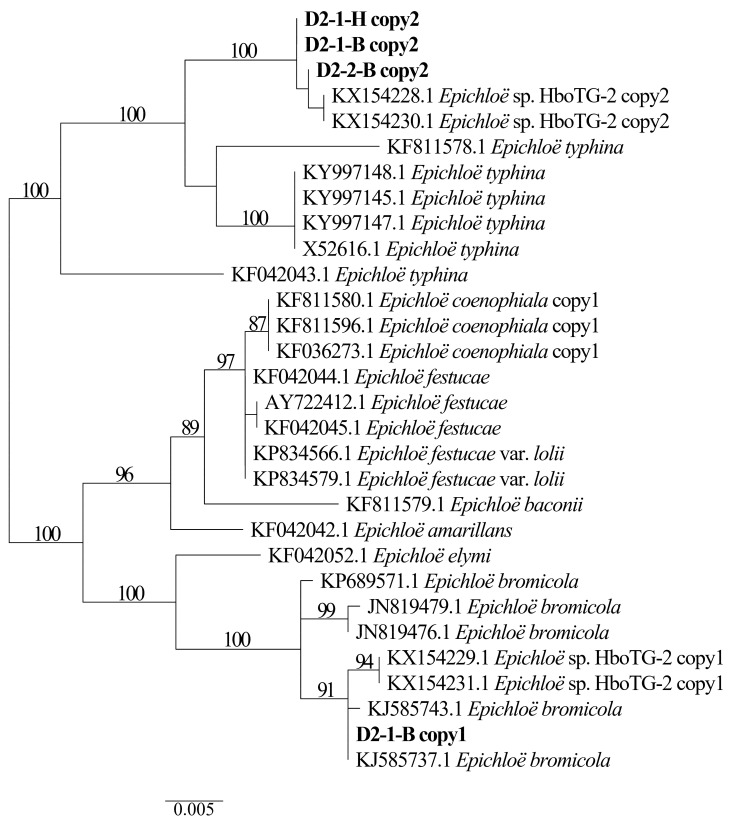
Molecular phylogeny derived from maximum likelihood (substitution model TNe + G4) analysis of the *tubB* gene from representative *Epichloë* species and copies of the hybridized endophytes isolated from *H. bogdanii*.

**Figure 4 jof-08-00928-f004:**
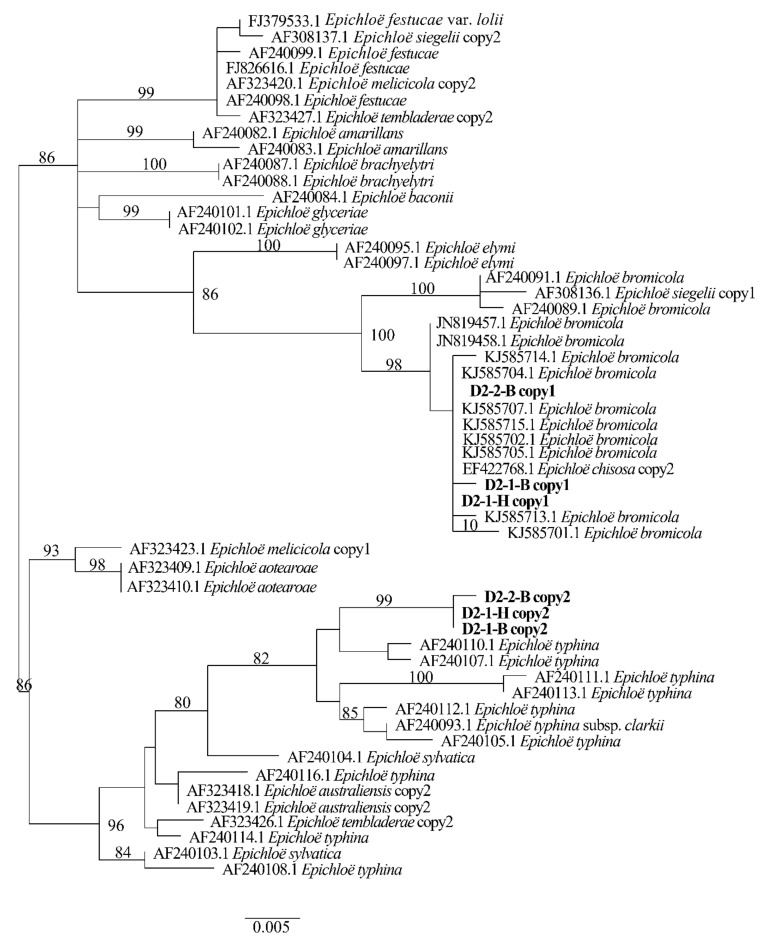
Molecular phylogeny derived from maximum likelihood (substitution model TIM2e + G4) analysis of the *actG* gene from representative *Epichloë* species and copies of the hybridized endophytes isolated from *H. bogdanii*.

**Figure 5 jof-08-00928-f005:**
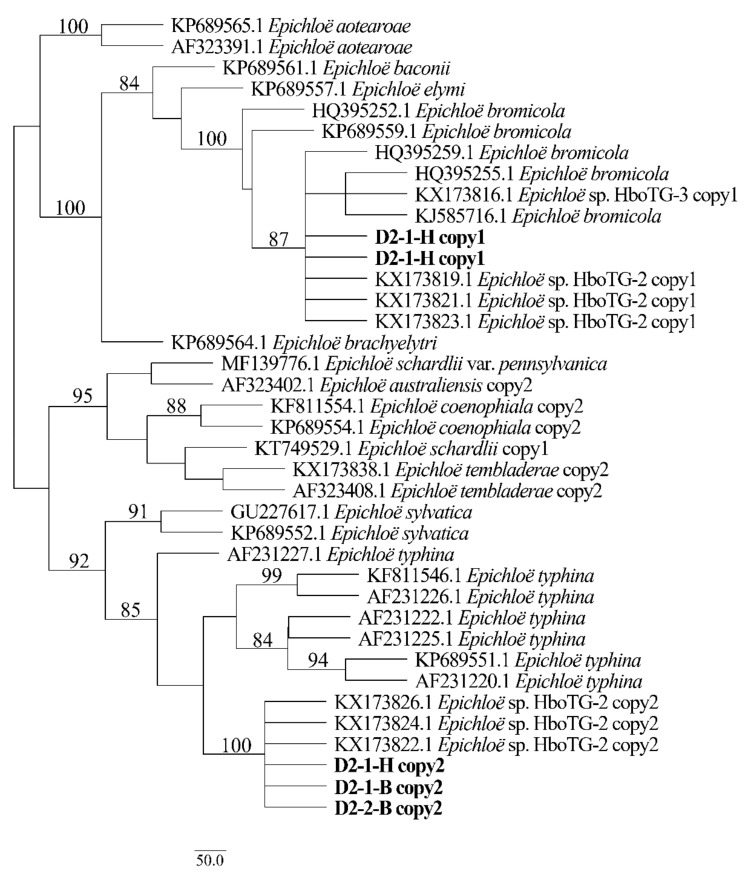
Phylogeny derived from maximum parsimony (MP) analysis of the *tefA* gene from representative *Epichloë* species and the copies of the hybridized endophytes isolated from *H. bogdanii*.

**Figure 6 jof-08-00928-f006:**
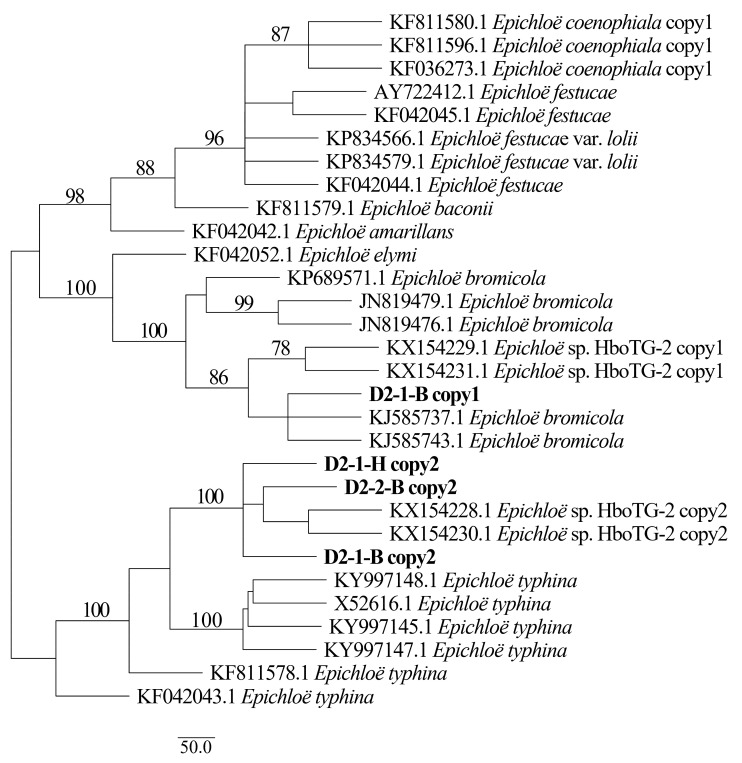
Phylogeny derived from maximum parsimony (MP) analysis of the *tubB* gene from representative *Epichloë* species and copies of the hybridized endophytes isolated from *H. bogdanii*.

**Figure 7 jof-08-00928-f007:**
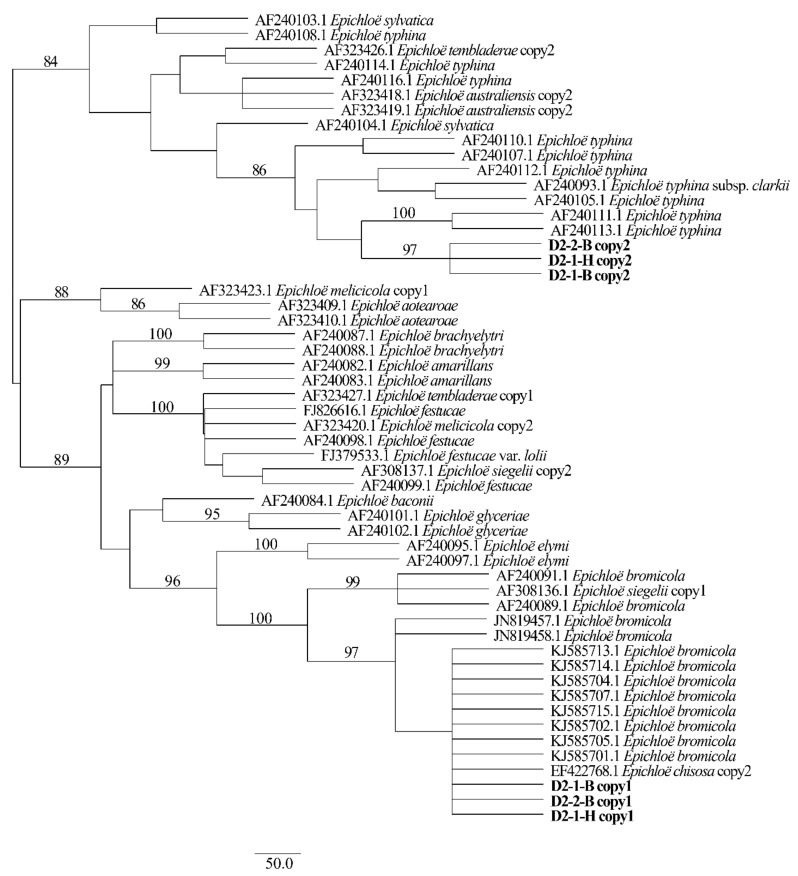
Phylogeny derived from maximum parsimony (MP) analysis of the *actG* gene from representative *Epichloë* species and the copies of the hybridized endophyte isolated from *H. bogdanii*.

## Data Availability

All data supporting the findings of this study are available within the paper.

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
