# Peer review of "Identification of Three Epichloë Endophytes from Hordeum bogdanii Wilensky in China"

_jof, 2022, doi:10.3390/jof8090928_

Round 1
Reviewer 1 Report
General comment:
In the submitted manuscript, three fungal endophytic isolates from the grass Hordeum bogdanii are analyzed by a four-gene phylogenetic analysis, morpho-physiological characteristics, and the presence/absence of alkaloid synthesis related genes. The obtained results indicate that the isolates are a new Epichloë interspecific hybrid species (HboTG-4). The submitted manuscript is well written, and the information provided is the required for an interspecific hybrid species in Epichloë. In general, all manuscript sections are clearly explained, and the arguments for the new interspecific hybrid species identification are properly described and discussed. The subject of the submitted manuscript is of interest for mycologists and phytopathologists. Thus, the manuscript is suitable to be published in the Journal of Fungi. Below are some specific comments for the authors' consideration.
Specific comments:
1. Despite your robust genetic analysis, I suggest modifying the sentence “Alkaloid chemotype analysis showed that…” (lines-18-19 in the Abstract) for something like “Alkaloid synthesis related genes analysis showed that…”. You do not perform chemical analysis as in previous works (e. g. Vikuk et al., 2019. Appl Environ Microbiol. 85(17): e00465-19).
2.Please provide geographical coordinates of the sampling area. This kind of information is relevant when new fungal species or strains from new geographical locations are described.
3. I strongly suggest performing the phylogenetic analysis in the Phylogeny.fr server. Although performing well for teaching and basic tasks, it is not recommended to use the MEGA software to generate a phylogenetic analysis for an advanced publication. Furthermore, you use an old version of the software. If you already use and know the friendly and robust Phylogeny.fr server, please reconstruct your phylogenies here. Another very good option superior to MEGA to perform a robust phylogenetic analysis is the IQ-TREE server (http://iqtree.cibiv.univie.ac.at/).
4. I suggest renaming Table 1 as “Morpho-physiological comparison…” The growth rate is not a morphological character.
5. I consider it necessary to conduct a statistical analysis (ANOVA, Tukey test) of your morphological data and growth rate of three studied three isolates in order to know if the differences are significant. If such differences are not significant, it is ok. But if such differences are significant what can be the probable explanations? In other words, intraspecific (intra-hybrid?) morpho-physiological variability has been previously reported in Epichloë spp.?
6. The figure 8 does not look like a ML reconstruction, but once you perform the new phylogenetic analysis in Phylogeny.fr or IQ-TREE, this can be reviewed.
Reviewer 2 Report
Please combine tefA, tubB, and actG genes sequence to reconstruct phylogenetic tree using maximum parsimony
Figures is not in printable quality. Also, some portions of the texts are losing their readability while sizing the image as per text area. Kindly provide better quality figure.
I recommend to update Introduction, and discussion sections with most recent review literature from2020-2022
English should be improved; grammar need for enhancement in many sentences and paragraphs.
Round 2
Reviewer 2 Report
Manuscript accepted in the current form